# Mapping of Major Land-Use Changes in the Kolleru Lake Freshwater Ecosystem by Using Landsat Satellite Images in Google Earth Engine

**Meena Kumari Kolli** [1,*], **Christian Opp** [1,*], **Daniel Karthe** [2,3] **and Michael Groll** [1]

1   Faculty of Geography, Philipps-Universität Marburg, Deutschhausstraße 10, 35037 Marburg, Germany; michael.groll@staff.uni-marburg.de
2   Environmental Engineering Section, German Mongolian Institute for Resources and Technology, Nalaikh, Ulaanbaatar 12800, Mongolia; karthe@gmit.edu.mn
3   Institute for Integrated Management of Material Fluxes and of Resources, United Nations University, 01067 Dresden, Germany
*   Correspondence: meenu.rgukt@gmail.com (M.K.K.); opp@staff.uni-marburg.de (C.O.)

**Abstract:** India's largest freshwater ecosystem of the Kolleru Lake has experienced severe threats by land-use changes, including the construction of illegal fishponds around the lake area over the past five decades. Despite efforts to protect and restore the lake and its riparian zones, environmental pressures have increased over time. The present study provides a synthesis of human activities through major land-use changes around Kolleru Lake both before and after restoration measures. For this purpose, archives of all Landsat imageries from the last three decades were used to detect land cover changes. Using the Google Earth Engine cloud platform, three different land-use scenarios were classified for the year before restoration (1999), for 2008 immediately after the restoration, and for 2018, i.e., the current situation of the lake one decade afterward. Additionally, the NDVI (Normalized Difference Vegetation Index) and NDWI (Normalized Difference Water Index) indices were used to identify land cover dynamics. The results show that the restoration was successful; consequently, after a decade, the lake was transformed into the previous state of restoration (i.e., 1999 situation). In 1999, 29.7% of the Kolleru Lake ecosystem was occupied by fishponds, and, after a decade of sustainable restoration, 27.7% of the area was fishponds, almost reaching the extent of the 1999 situation. On the one hand, aquaculture is one of the most promising sources of income, but there is also limited awareness of its negative environmental impacts among local residents. On the other hand, political commitment to protect the lake is weak, and integrated approaches considering all stakeholders are lacking. Nevertheless, alterations of land and water use, increasing nutrient concentrations, and sediment inputs from the lake basin have reached a level at which they threaten the biodiversity and functionality of India's largest wetland ecosystem to the degree that immediate action is necessary to prevent irreversible degradation.

**Keywords:** fishponds; Kolleru Lake; Google earth engine; eutrophication; land-use change

## 1. Introduction

Lakes are limited standing water bodies, in which changes in water, sediment, salinity, and pollutant inflow dynamics tend to have larger effects than in rivers [1]. In shallow lakes, changing environmental conditions can result in major regime shifts even within very limited time periods. For example, low water inflows or high sediment deliveries may threaten the existence of shallow water lakes within relatively short time periods, and pollutant concentrations can quickly reach levels that lead to massive changes in the lake ecosystem [2–4].

In most cases, lakes are degraded by processes that take place in their catchments, such as agricultural expansion, industrialization, urbanization, and deforestation [5–8]. Freshwater lakes are endangered through pollution from point and non-point sources. Industrial and domestic wastewater discharge, as well as polluted tributaries, constitute the dominant point sources, delivering a vast range of pollutants ranging from nutrients, toxic substances, and "emerging" pollutants such as pharmaceuticals and their metabolites or microplastic particles [9,10]. Non-point sources are mostly related to agricultural runoff [11–13]. In many lake basins, agricultural runoff is the key source of nutrient input, fostering eutrophication and algal blooms [14,15]. In fact, most lakes in the world suffer from eutrophication [16–18].

Even though there is considerable uncertainty about the total number, surface area, and water volume stored in lakes [19,20], several large lakes have significantly shrunk or even completely disappeared [21–23]. Identifying the causes of lake degradation is an important prerequisite for planning conservation and restoration measures [24–26]. Lake restoration is a time-consuming process that needs to consider both ecological aspects and human benefits from the lakes [27–30]. Studies in several parts of the world have investigated examples of more and less successful lake restoration, revealing that there is not one general blueprint for lake restoration but that it is necessary to consider the specific characteristics of lakes and their catchments [31–34].

Kolleru Lake, which is the largest standing surface water body in India, is a good example of a major lake that has come under increasing threats due to land-use change in its catchment, particularly due to growing aquaculture [35]. Problematic trends in the Kolleru Lake Basin include degradation of water quality, habitat losses, and a decrease in aquatic and avian biodiversity. These changes were induced by human activities and accelerated by climate change. The present study provides a synoptic assessment of land cover changes over the last three decades to compare the situation of the lake before and after restoration processes. This information provides a general understanding of lake degradation by major land-use changes, which can be useful for government management plans.

This study describes the application of remote sensing and GIS techniques for analyzing lake catchment [36,37] as an important prerequisite for planning restoration measures by visualizing, monitoring, and modeling temporal trends and intervention outcomes [38]. The methodological focus of this paper is on the utilization of Google Earth Engine (GEE), a cloud computing platform that comprises a large range of easily accessible data for visualization, mapping, analyzing, and modeling purposes [39,40]. In our case study, GEE was used to analyze the land-use conditions in the Kolleru Lake ecosystem before and after the restoration measures, with a specific focus on exploited areas of the lake through the application of multispectral images of several Landsat satellite generations. Even though a few studies on Kolleru Lake and its ecology already exist, they are dedicated to specific phenomena such as lake shrinkage [41] and specific aspects of water and sediment pollution [42,43]. The study presented here analyzed land-use conditions based on the computation of annual mean conditions using Landsat archives in GEE, with the ultimate goal of comparing the pre- and post-restoration state of the lake. The results not only add to the scientific knowledge of the Kolleru Lake system but also may be used as background for decision-making by the local government and other stakeholders.

## 2. Study Area

Kolleru Lake is located in the southeastern part of India, in the state of Andhra Pradesh, and close to the Bay of Bengal. Geographically, it is situated between 16°33′10″ and 16°47′44″ Northern latitude, and 80°4′5.5″ and 81°24′27.5″ Eastern longitude (Figure 1). It is the largest freshwater lake in India and forms one of the largest wetlands in Asia. All areas that are covered with water either permanently or seasonally are considered to be a part of Kolleru Lake. The average water covered area of the lake is 901 km$^2$ falling below 3.05 m contour level above MSL (mean sea level) contour level. Apart from that, the 1.52 m contour level of the lake area is designated as the Kolleru Wildlife Sanctuary (KWS) with a water spread area of 308 km$^2$, and the 0.91 m contour level with 245 km$^2$ is the actual lake area [44]. The average water depth is 1 m, and the maximum water depth reaches around 3 m

during the southwest monsoon period [45]. The lake is located in the tropical savanna climate zone (Aw, according to Koeppen) with an annual mean temperature of 28 °C and annual precipitation of 1094 mm, most of which falls during the months of the summer monsoon [44]. The lake receives the water from the seasonal rivers, namely Budameru, Tammileru, and Ramileru. Apart from this, 68 minor irrigation channels flow into the lake [42,46]. Kolleru Lake debouches into the Bay of Bengal through the meandering channel is called Upputeru [43].

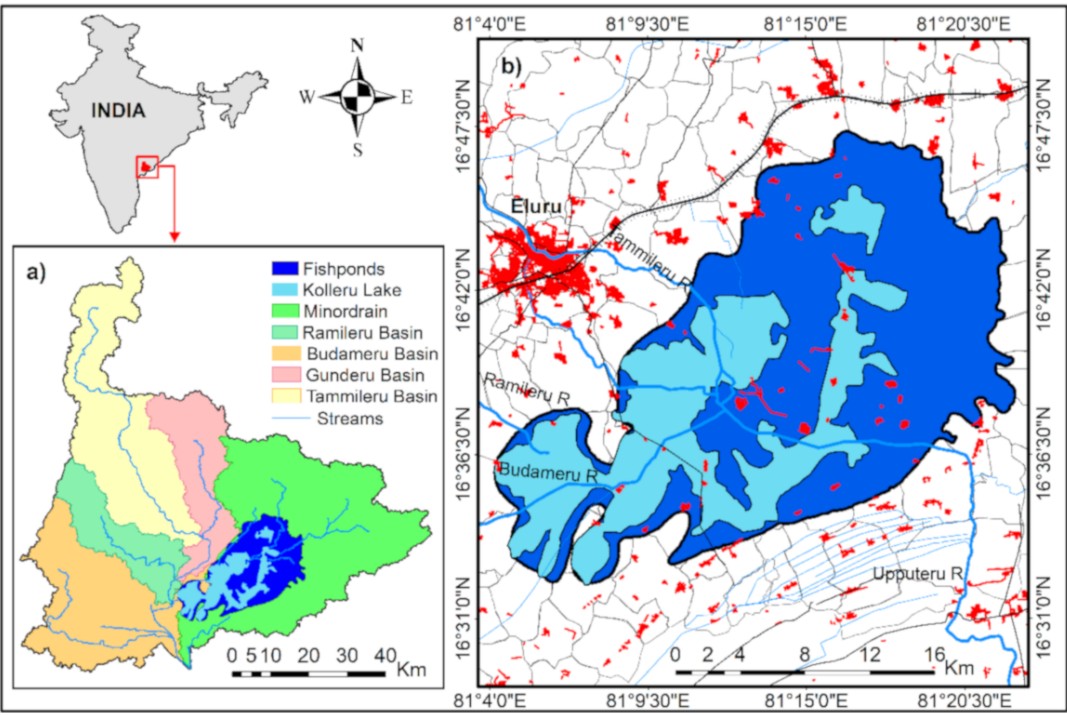

**Figure 1.** Location of the study site: (**a**) Kolleru Lake tributary river catchment; and (**b**) Kolleru Wildlife Sanctuary (in the dark blue background, i.e., fishponds occupied area).

The wetlands around Kolleru Lake form a hospitable environment for aquatic life, including an estimated 20 million resident and migratory birds which come to from northern Asia and Eastern Europe between October and March. The wetland has developed into a regional center for paddy cultivation fishing, aquaculture, and various recreational activities. However, a growing population with 0.3 million people live around Kolleru Lake, including 14,000 families who directly depend on lake resources. This has resulted in enormous environmental changes over the past four decades [41,47]. During the 1970s, the lake area was free from aquaculture pollution. When the state government allowed people to practice aquaculture and paddy cultivation within the lake region, significant parts of the lake area have developed into aquaculture ponds. Rao et al. [41] reported, based on a land-use classification for 2001, that Kolleru Lake is vanishing through large scale encroachment of illegal fishponds, and there was no trace of clear water in the lake. Thousands of illegal fishponds have encroached into the lake region so that open water areas are now difficult to identify in the satellite images. Since the restoration program was locally referred to as "Operation Kolleru", the following sections describe the situation prior to and after its implementation. To protect the lake ecosystem functions and services, "Operation Kolleru" was started in 2006 to dismantle the fishponds across this region, while being part of the "restoration of the lake" [47]. During that time, more than 1776 large tanks were destroyed, and 8.908 million m$^3$ of earth forming the tank bunds were removed [44]. Despite the dismantling of fishponds, water pollution caused by agricultural runoff resulted in proliferating weeds, i.e., water hyacinth (*Eichhornia crassipes*) and elephant grass (*Pennisetum purpureum*) (Figure 2a). The high rise embankments around the lake area, which obstructed the free flow of water, were dismantled during "Operation Kolleru" (Figure 2b). Therefore, the dry fishponds were from then

on considered as marshy areas (Figure 2c). Despite the changes that Operation Kolleru brought for fish farming, paddy cultivation remained a key economic activity of the 0.3 million local inhabitants of the Kolleru sanctuary (Figure 2d).

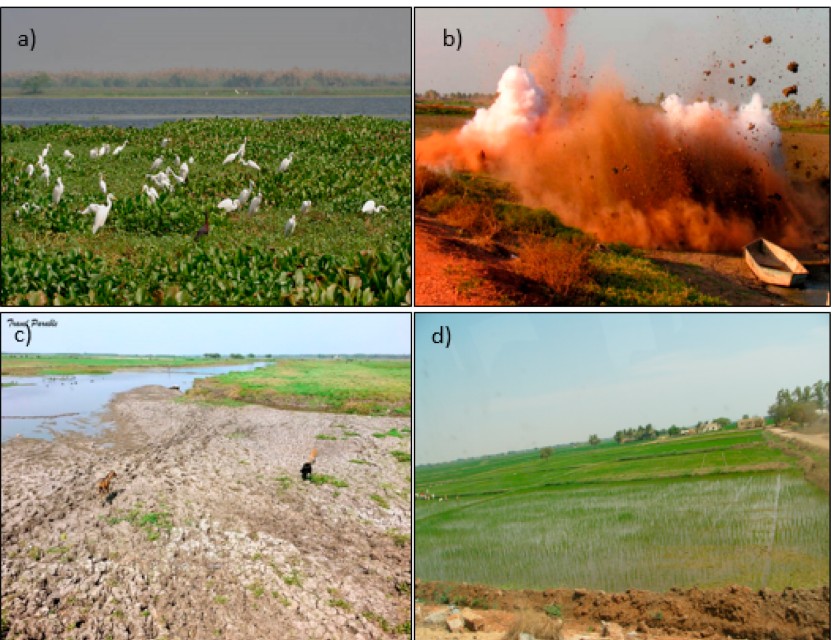

**Figure 2.** (**a**) Coverage of aquatic weeds (Photo by J.M.Garg, 9 October 2008); (**b**) demolition of illegal fishponds by the "Operation Kolleru" program during 2005–2006 [48]; (**c**) marshy areas (Google image); and (**d**) paddy crops grown in the lake catchment area (Photo by Opp, 2010).

Several studies showed that the development of large-scale aquaculture has a detrimental effect on the Kolleru Lake ecosystem [35,48–51]. There is also growing evidence that, beyond nutrient and sediment input, the waters and lake sediments of Kolleru Lake are increasingly polluted by heavy metals [52–54], pesticides [55], and PAHs [56]. These pollutants have entered the aquatic food web, sometimes leading to critical levels of bioaccumulation in fish reared in the aquafarms [43,53,56,57]. Diffuse pollution from agriculture runoff is a major threat to water quality. However, it is an essential contribution to the total loads of nitrate-nitrogen ($NO_3\_N$), and total phosphorus (TP) is explained in our previous studies [47].

## 3. Data and Methods

Google Earth Engine (GEE) is a platform storing a large volume of Earth observation data over the past four decades, including satellite imagery from frequently used platforms such as Landsat, Sentinel, MODIS, and other geospatial information such as climate and population data [39,40]. In the case of Landsat, GEE provides access to images sourced from the United States Geological Survey (USGS). For land surface cover information, data range from Landsat 1 Multispectral Scanner (MSS) data (1972–1978) to the current generation Landsat-8 Operational Land Imager (OLI) (from 2013) [58].

Data for the identification of major land-use changes in the Kolleru freshwater ecosystem over the last three decades were derived from the Landsat archives available in GEE (Figure 3). Landsat-7 ETM+ images for 1999 and 2008 and Landsat-8 OLI TIRS images for 2018 were used to classify the landcover in the Kolleru freshwater ecosystem. All bands with 30-m resolution were used. The images were orthorectified and attributed to top-of-atmosphere reflectance with minimum cloud cover as primary data input. Before importing the Landsat data into the GEE, the first step involved in the removal of the cloud shadow and cloud cover, respectively, since GEE did not integrate the images with atmospheric correction already performed. In a second step, the Normalized Difference Vegetation Index (NDVI) and Normalized Difference Water Index (NDWI) indices were calculated as annual means.

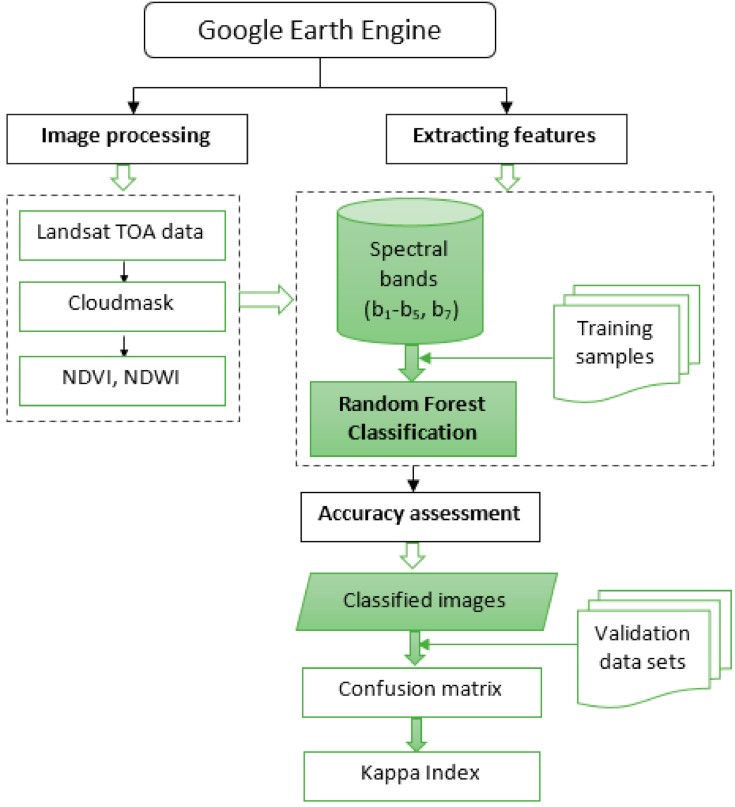

**Figure 3.** Methodology flowchart used for image classification.

GEE stores all individual datasets; for this study, those images that cover the entire study area up to 1.52 m contour level were used. This resulted in different numbers of usable Landsat images: for 1999 in total 3 images, for 2008 in total 23 images, and for 2018 in total 43 images. For each year, Landsat data were aggregated into a single image by applying the median function to receive a composite image at 30-m spatial resolution. However, it calculated each pixel with a median value for the entire stack of images. Therefore, the result is having one image representing the entire image collection. There are seven bands in Landsat images excluding the thermal bands; in addition, we added the NDVI and NDWI indices for better land use mapping accuracy.

The integration of NDVI and NDWI time-series and textural features help to better differentiate the water body and vegetation to achieve the maximum classification accuracy (Figure 4) [59]. The combination of machine learning algorithms and remote sensing data is more efficient, simple, and useful for mapping wetland dynamics [60]. To produce a land-use classification, we used 70 training sites from high-resolution Google Earth images for six different land-use classes. The number of samples was determined for each land-use type according to the proportion of its area.

Furthermore, training sample data were integrated into the GEE as a featured class table. We used spectral bands 1–7 of the Landsat images and added information on NDVI (Equation (1)) and NDWI (Equation (2)) to achieve the best possible classification accuracy.

$$\text{NDVI} = \frac{\rho_{\text{NIR, i}} - \rho_{\text{RED, i}}}{\rho_{\text{NIR, i}} + \rho_{\text{RED,i}}} \tag{1}$$

$$\text{NDWI} = \frac{\{\rho(0.86 \ \mu\text{m}) - \rho(1.24 \ \mu\text{m})\}}{\{\rho(0.86 \ \mu\text{m}) + \rho(1.24 \ \mu\text{m})\}} \tag{2}$$

where $\rho_{\text{RED}}$ is the radiance of the red channel (i.e., near 0.66 μm), $\rho_{\text{NIR}}$ is the radiance of near-IR wavelength (i.e., near 0.86 μm), $\rho(\lambda)$ is apparent reflectance, and $\lambda$ is the spectral band wavelength. It is derived from a Near-IR band and a second IR band [61].

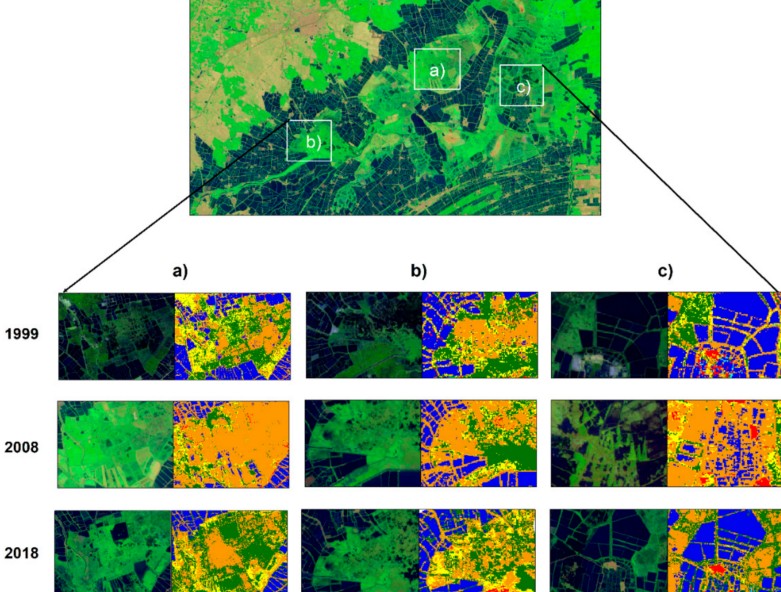

**Figure 4.** The 30-m Landsat derived land-use classes for the years 1999, 2008, and 2018 for the Kolleru Wildlife Ecosystem.

GEE currently provides the machine learning algorithm, such as the Random Forest algorithm, which was used to classify the Landsat image in this study [62]. The Random Forest algorithm is faster and more robust than other regression models where it merges the output of multiple decision trees to generate the final output. Besides, NDVI and NDWI composite time series throughout the year were used as additional data for phenomenal characteristics. Finally, the study area was classified into six different classes: built-up land, paddy fields, weed infests, marshy land, the open lake area, and fishponds. The open lake area was considered a non-wetland class in this study, referring to only such parts of the lake surface that were covered by neither natural or manmade marshland nor anthropogenic land use. The classification system was specific to the first-level algorithm of the Random Forest classification scheme. Further accuracy assessment analysis was performed to quantify the accuracy of classified images. A confusion matrix is an Earth Engine built algorithm to check the accuracy of the detailed map and further to assess the accuracy of the classification in the earth engine. We used training datasets that were split into both training and validation in GEE to determine the overall accuracy and Kappa coefficient. Kappa analysis is a discrete multivariate technique to measure image accuracy. The kappa index ranges between −1 and +1, with a level of +1 describing a perfect agreement [63].

## 4. Results and Discussion

The results demonstrate that the Kolleru Lake ecosystem was impacted by both government decisions (to introduce aquaculture and to restore the lake, respectively) and their implementation by the local population. In this study, the land use classes were categorized into two types: wetland classes of fishponds, weed infest, paddy fields, and marshy areas and non-wetland classes of built-up land and lake open areas.

### 4.1. Pre-Operation Kolleru

In 1999, most of the Kolleru wetland area was covered by fishponds, followed by paddy cultivation, settlements, weed infests, marshy areas, and open lake areas. Figure 5 illustrates the rapid expansion of fishponds within the Kolleru Lake 1.52 m contour level, which substantially disturbed the lake ecosystem in 1999. Approximately 146.4 km$^2$ (29.7%) of the lake area was occupied by fishponds of the lake sanctuary in 1999. The weed (91 km$^2$) and marshy areas (77 km$^2$) were water-covered

during the monsoon period, but turned into moist land during the summer period, with weeds proliferating. The paddy fields (106.4 km$^2$) were the second dominant land use in 1999, and most of the paddy areas were already converted to the fishponds to minimize the impacts of monsoonal floods. The non-wetland classes covered less than 4% of the total area, beyond which the pattern of open lake areas was difficult to identify in the classified image.

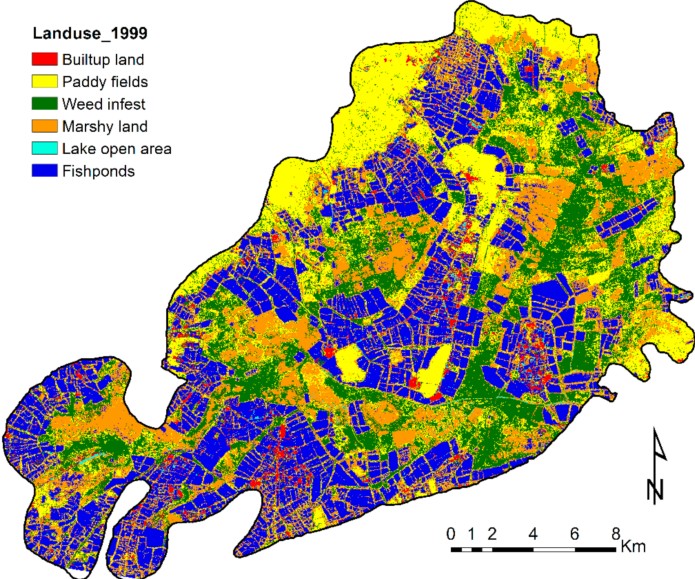

**Figure 5.** The 1999 land-use map of the Kolleru Lake wetland based on supervised classification.

In addition to information that is directly visible in the land use maps, it was reported that migratory birds were slowly disappearing due to aquaculture-related pollution in this region [44]. Moreover, during the monsoon heavy flooding period, all the riparian villages were severely affected due to the submersion of crops, a problem that was aggravated by the fishponds [48].

### 4.2. Post-"Operation Kolleru"

After the successful "Operation Kolleru" restoration program, the maximum percentage of the lake area (71.2%) was free from aquaculture pollution. Figure 6 illustrates that most of the former fishpond areas were categorized into the marshy landcover class in 2008, which is related to the government's ban on aquaculture in the lake region. Marshy areas became the dominant landcover class at an area of 294 km$^2$, whereas 76.4 km$^2$ of fishponds remained, and paddy-fields and weed-infested areas covered another 56.2 and 55.2 km$^2$, respectively. Settlements cover less than 4% of the total area. In particular, the results show that the areas under aquaculture are reduced by approximately half due to the restoration program.

Unfortunately, the regional government did not sustain its activities after Operation Kolleru, and local people once again started encroaching the area. Therefore, a considerable number of fishponds reappeared between 2008 and 2018.

Figure 7 shows that, in 2018, fishponds occupied 136 km$^2$ of the area, which is about 2% less than in 1999, and thus indicates that, after restoration measures, the continuous monitoring and regulation of illegal fishpond growth was not sufficiently implemented by the government. Marshy areas made up 166 km$^2$, weed infests areas covered 152 km$^2$, paddy fields covered 28.7 km$^2$, and the settlements occupied the rest of the area. Aquaculture thus continues to be one of the main threats to Kolleru Lake's ecosystem. In 2018, 27.9% of the wildlife sanctuary was occupied by fishponds, which have not only replaced other land cover types but are also an important pollution source for the entire lake. Because the regulation of fishponds is laced up with sociopolitical concerns, local governments need to consider tradeoffs between ecosystem conservation and local economic development.

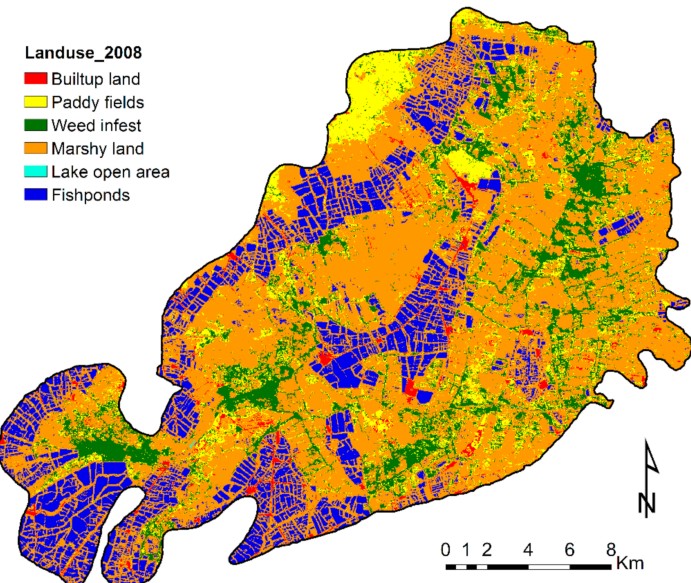

**Figure 6.** The 2008 land-use map of the Kolleru Lake wetland based on supervised classification.

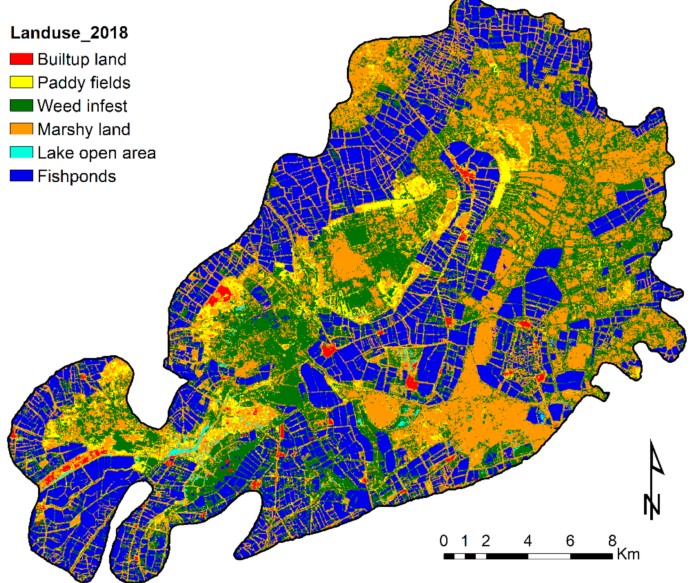

**Figure 7.** The 2018 land-use map of the Kolleru Lake wetland based on supervised classification.

*4.3. Accuracy Assessment*

A standard accuracy test was performed to quantify the accuracy of each classified image. Eighty percent of data were used for training using the random forest model, and 20% for validation using a confusion matrix.

The statistics of accuracy (overall accuracy, producer's accuracy, consumer's accuracy, and Kappa coefficient) that were achieved by the machine learning algorithm in GEE applied to 1999, 2008, and 2018 land-use classifications are presented in Table 1. The mean overall accuracy and Kappa coefficient for 1999 were 88.42% and 0.84, respectively; for 2008, they were 95.99% and 0.94, respectively; and for 2018, they were 88% and 0.82, respectively. Overall, the highest producer's and consumer's accuracies were achieved for wetland classes as compared to non-wetland classes. However, the accuracy of open lake areas was much worse than for other land use classes. The very few pixel values that were generated by the Landsat images implies very few training samples were chosen in GEE. The Random Forest

algorithm was not able to detect the open lake areas in this study. The data uncertainty present in this particular land-use class reflects the relatively low producer's and consumer's accuracies.

**Table 1.** Accuracy assessment test of the land use types for 1999, 2008, and 2018.

| Accuracy Assessment (1999) | | | | | | | |
|---|---|---|---|---|---|---|---|
| **Types** | **Urban** | **Paddy** | **Weed** | **Marshy Land** | **Lake Open Area** | **Fishponds** | **Producer's Accuracy** |
| Urban | **126** | 6 | 0 | 5 | 2 | 8 | 85.71 |
| Paddy | 3 | **2595** | 203 | 44 | 2 | 3 | 91.05 |
| Weed | 0 | 231 | **1553** | 65 | 3 | 0 | 83.85 |
| Marshy land | 0 | 67 | 71 | **1283** | 3 | 46 | 87.27 |
| Lake open area | 7 | 8 | 11 | 63 | **21** | 21 | 16.03 |
| Fishponds | 2 | 3 | 0 | 52 | 4 | **1548** | 96.21 |
| Consumers accuracy | 91.30 | 89.17 | 84.49 | 84.85 | 60.01 | 95.20 | |
| **Overall accuracy: 88.42%, Kappa coefficient: 0.84** | | | | | | | |
| Accuracy Assessment (2008) | | | | | | | |
| **Types** | **Urban** | **Paddy** | **Weed** | **Marshy Land** | **Lake Open Area** | **Fishponds** | **Producer's Accuracy** |
| Urban | **76** | 2 | 0 | 0 | 0 | 0 | 97.43 |
| Paddy | 6 | **758** | 38 | 20 | 0 | 1 | 92.10 |
| Weed | 0 | 66 | **1007** | 23 | 0 | 0 | 91.87 |
| Marshy land | 0 | 29 | 30 | **2069** | 1 | 19 | 96.32 |
| Lake open area | 0 | 0 | 0 | 12 | **0** | 0 | 0 |
| Fishponds | 0 | 0 | 0 | 16 | 0 | **2393** | 99.33 |
| Consumers accuracy | 92.68 | 88.65 | 93.67 | 96.68 | 0 | 99.17 | |
| **Overall accuracy: 95.99%, Kappa coefficient: 0.94** | | | | | | | |
| Accuracy Assessment (2018) | | | | | | | |
| **Types** | **Urban** | **Paddy** | **Weed** | **Marshy Land** | **Lake Open Area** | **Fishponds** | **Producer's Accuracy** |
| **Urban** | **117** | 2 | 1 | 25 | 0 | 0 | 80.68 |
| **Paddy** | 3 | **241** | 24 | 44 | 0 | 0 | 77.24 |
| **Weed** | 0 | 12 | **1005** | 107 | 2 | 4 | 88.93 |
| **Marshy land** | 12 | 32 | 127 | **1386** | 2 | 10 | 88.33 |
| **Lake open area** | 1 | 1 | 4 | 16 | **32** | 0 | 59.25 |
| **Fishponds** | 1 | 0 | 7 | 12 | 1 | **519** | 96.11 |
| **Consumers accuracy** | 87.31 | 83.68 | 86.04 | 87.16 | 86.48 | 97.37 | |
| **Overall accuracy: 88%, Kappa coefficient: 0.82** | | | | | | | |

### 4.4. Analysis of Land-Use Changes for Three Decades

The Kolleru Lake wetland has been subjected to massive anthropogenic degradation since the 1990s. Figure 8 shows the land use analysis of the Kolleru Lake ecosystem (1.52 m contour level) according to the Random Forest algorithm based on the machine learning approach. During 1999, the highest percentage of land was occupied by fishponds, around 29.7%, followed by paddy fields (23.6%), marshy land (22.1%), and weed-infested areas (20%). The built-up areas and open lake areas were less than 5%. Considerable changes were brought by "Operation Kolleru" in 2006, during which a considerable number of fishponds were dismantled within the lake region, which was turned into "dry fishponds". These dry fishponds were classified as marshy land, which is due to regular inundations of the former ponds during and after the monsoon period. In 2008, marshy land thus became the dominant land-use class, covering about 59.8% of the area. The area covered by fishponds areas reduced to 15.5%, whereas paddy fields and weed infests occupied 11.4% and 11.2% of the area, respectively. Whereas the satellite images of 2008 document the success of "Operation Kolleru",

illegal encroachments on dry fishpond areas resumed in the following years. The 2018 land use classification reveals that fishponds once again occupied 27.7% of the surface area, which is very close to the 29.7% observed in 1999 before restoration. At the same time, marshy land, weed infests, and paddy fields made up 33.7%, 30.9%, and 5.84% of the wetland, respectively. The lake's self-purification capacity reduced, exacerbating the levels of nutrient pollution and weed proliferation over time [47]. Even though the government of India took considerable efforts to restore Kolleru Lake in the early 21st century, "Operation Kolleru" turned out to be only a short-term success due to a lack of integrated approaches and sustained regulation of aquaculture, as well as a lack of scientific information and continuous monitoring for policy planning.

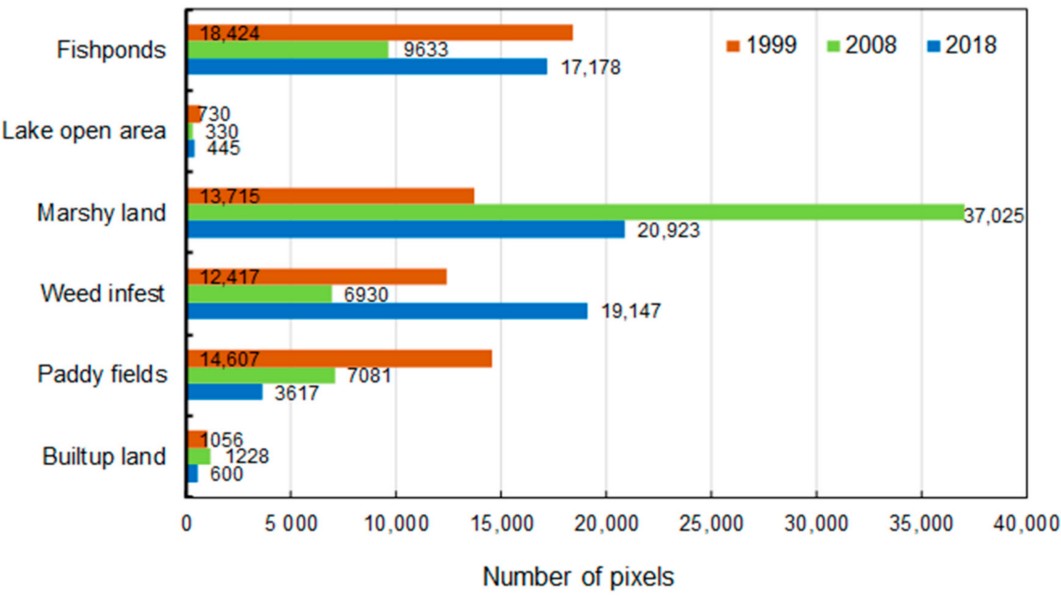

**Figure 8.** Land use change in the Kolleru Lake wetland between 1999 and 2018 (Figure Updated).

## 5. Conclusions

As India's largest freshwater lake and one of the largest wetland areas in Asia, Kolleru Lake has been declared a RAMSAR wetland site of international importance. Despite this, the lake and its surrounding wetland experienced a massive deterioration of its ecosystem, primarily driven by strong growth in aquaculture. Operation Kolleru, which was launched by the Indian government in 2006, managed to reverse some of the land-use changes, resulting in the decommissioning of approximately half of the fishponds. However, our results show that it was not sustainable in the long term. Even though it is understandable that aquaculture is an important source of income among the local population, there is a relatively urgent need for further action to limit its impacts on the lake and wetland ecosystem. At the same time, it is clear that, in a densely populated country such as India, strict conservation (without any human use) is quite illusionary for a large-scale wetland region such as that of Kolleru Lake. At this stage, the pragmatic approach proposed by the RAMSAR convention and its updates, the so-called "wise use concept", might be the most suitable solution strategy by looking at the aquatic ecosystem in an integrated way. As already proposed for Kolleru Lake [64], considering both environmental aspects and regional socioeconomic concerns has the greatest likelihood of succeeding in the long-term. One very important prerequisite for such a management approach is the scientific monitoring of the wetland's health. The remote-sensing based land use classifications presented here are one important contribution but will require future updates to assess the outcomes of management interventions and other trends within the wetland.

**Author Contributions:** Conceptualization, D.K.; Formal analysis, M.G.; Funding acquisition, M.K.K.; Methodology, M.K.K.; Project administration, C.O., D.K. and M.G.; Supervision, C.O.; Visualization, D.K.; Writing—original draft, M.K.K.; Writing—review & editing, C.O. and D.K. All authors have read and agreed to the published version of the manuscript.

**Funding:** This research received no external funding.

**Acknowledgments:** The authors would like to thank anonymous referee for their valuable comments.

**Conflicts of Interest:** The authors declare no conflict of interest.

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
