# Peer review of "Mapping of Major Land-Use Changes in the Kolleru Lake Freshwater Ecosystem by Using Landsat Satellite Images in Google Earth Engine"

_water, doi:10.3390/w12092493_

Round 1

Reviewer 1 Report

Review comments

Section 3 (Data and Methods) -The assessment should be complemented using alternative methods to support land use classification, solely based on remote-sensing (vegetation & water Index) or verify the analysis. List of papers suggested below, for alternative approaches to consider, e.g. to detect changes, map trends, and quantify differences on the land use.

Line 147-149 - the statement is not clear. 

Section 4.4 (Line 251-272) Analysis of land-use changes for three decades

The land-use change (Figure 8) accuracy/uncertainty is not covered.

Suggested references:

  • Yihdego, Y. and Webb, J.A., 2011. “Modeling of bore hydrograph to determine the impact of climate and land use change in a temperate subhumid region of south-eastern Australia. Hydrogeology Journal, Volume 19, Issue 4, 2011, page 877-887. http://link.springer.com/article/10.1007/s10040-011-0726-1
  • Yihdego, Y. and Webb, J.A., 2017 "Assessment of wetland hydrological dynamics in a modified catchment basin: case of Lake Buninjon, Victoria, Australia". Water Environmental Research Journal Volume 89(2): 144-154. doi: 10.2175/106143016X14798353399331. https://doi.org/10.2175/106143016X14798353399331
  • Yihdego, Y. and Webb, J.A., 2008. Modelling of Seasonal and Long-term Trends in Lake Salinity in Southwestern Victoria, Australia. Proceedings of Water Down Under April 2008, 994-1000, Adelaide. Engineers Australia.  Casual production, 2008: 994-1000. ISBN: 0858257351.
  • Yihdego, Y and Webb, J.A., 2013. “An empirical water budget model as a tool to identify the impact of land-use change in stream flow in southeastern Australia,” Water Resources Management Journal, 27 (14), 4941-4958. http://dx.doi.org/1007/s11269-013-0449-2

Author Response

Author Response to the Reviewer:

Line 147-149: The sentence added with appropriate information.

Section 4.4 (Line 251-272): Analysis of the three decades was explained clearly.

Figure8: Updated.

Reviewer 2 Report

Review of

Mapping of major land-use changes in the Kolleru Lake freshwater ecosystem by using Landsat 2 satellite images in Google Earth Engine

Meena Kumari Kolli et al.

Overall this is an excellent paper of good technical, writing, and scientific value. It demonstrates excellent use of GEE for wholistic analysis of a wetland ecosystem in India and provides excellent real-world implications for usage. I recommend publishing after only minor revisions listed below.

Line17 : I would add “environmental” to describe type of pressures

 Despite efforts to protect and restore the lake and its riparian zones,  environmental pressures have 17 increased over time.

Line 81: extra space between the and    South-Eastern

Line 140: I think “median” not “medium” (or mean) should be used here?

Clarify this statement to describe that this study used the Random Forest Algorithm.

Line 163 “such as the Random Forest algorithm 163 was used for supervised classification to classify the Landsat image”

Line 171: Please clarify and provide more details about this sentence and material. There is not enough information for a user to reproduce what was done (e.g., what is a Kappa coefficient, what does it tell us?)

“A confusion matrix is an Earth Engine built algorithm to 171 check the accuracy of the detailed map and further to assess the accuracy of the classification in the 172 earth engine. We used training data sets that were split into both training and validation in GEE to 173 determine the overall accuracy and Kappa coefficient.”

Line 201: What % of the wetland area was covered by fish pond? I think giving a % of the total area would be effective. Figure 5 -- Approximately 146.4 km2 of the lake area was occupied by fish ponds

While there was a 7% reduction in fish ponds from 1999 to 2018, I think you should also point out that there has been a dramatic increase once again in fish ponds after the dramatic decrease during the operation in 2008. What will the future bring? Are more operations planned?

Line 221: This is a very telling statement that should be more highlighted in the abstract, intro, and conclusion in my opinion: “Unfortunately, the regional government did not sustain its activities after operation Kolleru, and 221 local people once again started encroaching the area. Therefore, a considerable number of fishponds 222 reappeared between 2008 and 2018.”

The first link in the google earth engine did not work, please verify that the data is there. The GEE is a very powerful way to increase scientific use of large datasets and the authors have done an excellent job. I would recommend adding a few more sentences and references of other studies in the past year that have used GEE in the introduction since this is an emerging resource of high importance (GEE).

Author Response

All the minor corrections corresponding to the reviewer feedback have been sentenced and updated. 

Reviewer 3 Report

General notes

Please check the English of the text

The descriptions of the derived maps are very weak. Please, 1) describe the areal extension of certain land-use types; 2) give a description on the spatial distribution of each land-use types (e.g. where was the lake in 1999?) 3) indicate in a map those areas where the same land-use remained, and where the land was transformed into something else (continuity map). Please, analyse the changes (in relation to closeness of urban areas, relief etc.)

Abstract

  1. 15: „The Kolleru Lake” Please, define where is it (country, area)
  2. 18: „ land-use changes” I am confused here, as before you mentioned just the illegal fish pond construction as the sign of land-use change. Do you plan to map other changes too (change in agricultural land area)? Please, add a half sentence on that!

Introduction

  1. 44 (and at other places too) „ [1] [2] [3]” When you cite 2 or more papers, please change the format to [1-3] or [1, 11 and 14]. Please, check the whole text and correct it.
  2. 64: please, describe the environmental problem and threats regarding the lake. What are your research goals and aims exactly? Please specify them here, before you would introduce the way of solution.

Study area

Fig. 1. Usually the contour lines are indicated by brown or black lines, but never by blue because that is colour is reserved for waters. Please, change it. What do you mean on sanctuary? If it is a nature reserve, it should have distinct border, but I do not see it on Fig. 1b. Please, indicate it applying some colour-line.

Please, use SI units. Why do you use feet?

  1. 93: “irrigation channels flow into the lake” usually drainage channels can drain water into the lake, while irrigation channels drain water out of lakes. I do not understand here the direction of the canals.
  2. 103 „growing population” How many people lives here? How many are dependent on the lake?
  3. 108: „Kolleru Lake is vanishing through large scale encroachment of illegal fishponds, and there was no trace of clear water in the lake”. As it is the key momentum of your study, please, describe in greater detail: why is the lake declining if there are fishponds around? (I understood it only later on, that the lake area was transformed into fish ponds, so it is not clearly described here!!) What does it mean no trace of clear water?

You should describe the environmental problem and its causes in greater detail here, otherwise your study is not prepared/founded properly.

Methods

Fig. 3. For better visualisation, please use white rectangular and letters on the first figure, and leave some space between the a-b-c pictures, as they are not connected in space.

  1. 169: „the open lake area, and fishponds” How do you distinct them? What is their definition, and how could you separate the lake and a fishpond (made of lake)? It is a crucial question, as the lake itself seems to be disappearing. What is the minimum size of fish-pond you can identify in this way?

Usually the Gini-index is applied to see the efficiency of the classification. Please, consider to use it.

Results

  1. 176- 193: These paragraphs are definitely not based on your data, it would fit more to the introduction or to the study area. Please, replace it.
  2. 182: „89.08 lakh (1 lakh = 0.1 million) cubic meters” please, express it in m3, and if you want, in bracket you can express t in lakh too, but as a reader, I do not want to make calculations!
  3. 185-193. This paragraph is very strange: its English is bad, and the facts you list here seem to be randomly mentioned, there is no clear line of idea.
  4. 197- I suppose, these are your results, but now they are mixed with other information, so I am not sure. Please emphasise your contribution. The description of the derived map is very weak. Please, 1) describe the areal extension of certain land-use types; 2) give a description on the spatial distribution of each land-use types (where was the lake in 1999?)
  5. 211: „maximum percentage of the lake area.” What do you mean on it? 100% of the lake area??

Reference list

Please check the format requirements of the journal (names, order of years etc.), and definitely the font size and type, and the line spacing.

Round 2

Reviewer 3 Report

Dear Authors,

thank you for the detailed correction of the text!

I hope your article will attract the interest of the scientific community!
All the best!